# Facile Fabrication of Fluorine-Free, Anti-Icing, and Multifunctional Superhydrophobic Surface on Wood Substrates

**DOI:** 10.3390/polym14101953

**Published:** 2022-05-11

**Authors:** Mengting Cao, Mingwei Tang, Wensheng Lin, Zehao Ding, Shuang Cai, Hanxian Chen, Xinxiang Zhang

**Affiliations:** 1College of Materials Engineering, Fujian Agriculture and Forestry University, Fuzhou 350108, China; 18271421297@163.com (M.C.); wensheng0817@163.com (W.L.); a1787844623a@163.com (Z.D.); hanxian1229@163.com (H.C.); 2Department of Chemical Engineering and Food Science, Hubei University of Arts and Science, 296 Longzhong Road, Xiangyang 441053, China; tangmingwei2022@163.com

**Keywords:** wood, multifunctionality, poly(methylhydrogen)siloxane (PMHS), wettability, anti-icing

## Abstract

Building superhydrophobic protective layers on the wood substrates is promising in terms of endowing them with multiple functions, including water-repellent, self-cleaning, anti-icing functions. In this study, multifunctional superhydrophobic wood was successfully fabricated by introducing SiO_2_ sol and superhydrophobic powder (PMHOS). The SiO_2_ sol was prepared using tetraethoxysilane as a precursor and ethanol was used as the dispersant. The PMHOS was synthesized using poly(methylhydrogen)siloxane (PMHS) and ethanol. As a result, the obtained superhydrophobic wood had a water contact angle (WCA) of 156° and a sliding angle (SA) of 6° at room temperature. The obtained superhydrophobic wood exhibited excellent repellency toward common liquid (milk, soy sauce, juice, and coffee). The superhydrophobic layer on the wood surface also exhibited good durability after a series of mechanical damages, including finger wiping, tape peeling, knife scratching, and sandpaper abrasion. In addition, the obtained superhydrophobic wood showed excellent anti-icing properties.

## 1. Introduction

As a natural, renewable, and environmentally friendly material, wood is widely used in daily life, such as in wooden houses, wooden boats, and wooden carvings [1,2]. However, wood is a polymer material composed of cellulose, hemicellulose, and lignin. It has a rich pore structure and contains many hydrophilic hydroxyl groups [3,4]. Therefore, wood has a strong affinity for water absorption, resulting in mildew, cracking, and discoloration, all of which severely shorten the wood’s service life [2,5]. Inspired by the excellent water repellency of some plants and animals in nature, such as lotus leaves, rice leaves, butterfly wings, and water strider legs [6,7,8,9], the construction of superhydrophobic coatings on wood substrates is considered to be one of the most effective ways to reduce wood’s wettability [10,11,12]. Here, the worse the wettability of the material is, the better the water repellency will be. Therefore, by building a superhydrophobic film on wood substrates, the service life of the wood can be prolonged. 

A surface with a WCA greater than 150° and a sliding angle (SA) less than 10° is a superhydrophobic surface [13,14,15]. In recent years, superhydrophobic surfaces have attracted attention due to their many excellent properties, such as anti-icing [16,17], anti-fogging [18,19], anti-corrosion [20], self-cleaning [19,21], drag-reduction, and oil/water separation properties [22,23,24]. According to previous reports [25,26], the construction of superhydrophobic surfaces can be achieved by increasing the surface roughness or reducing the surface energy. Numerous studies have reported various methods for constructing superhydrophobic surfaces on wood substrates, including the sol–gel process [27], hydrothermal method [28,29], plasma treatments [30], CVD [31], and LBL [32]. For example, Tu et al. fabricated mechanically robust superhydrophobic layers on a wood surface using epoxy resin, silica, and fluorinated alkylsilane (FAS). The modified wood possessed excellent durability against water spray impact and UV radiation [33]. Kong et al. successfully fabricated a ZnO coating on a wood surface via a hydrothermal process. The obtained modified wood possessed excellent flame retardancy and enhanced photostability [25]. Wang et al. obtained a superhydrophobic wood by using fluorine-containing reagents. The wettability of the wood surface was changed from hydrophilic into water repellent with a WCA of 164° and an SA of less than 3° [27]. Obviously, most superhydrophobic woods have been prepared by using expensive and toxic chemicals. Some modification methods are only suitable for modifying small pieces of wood (such as the hydrothermal method), which shortens their application field. A large number of research reports were focused on the construction of superhydrophobic coatings on the surface of wood substrates; however, the resulting superhydrophobic surfaces have poor durability. Therefore, we need to find a simple, low-cost, and environmentally friendly method for constructing durable superhydrophobic coatings on wood surfaces.

In this study, we proposed a method for preparing superhydrophobic wood with a good durability and an anti-icing property. The functionalized wood was readily obtained by dipping the wood into a SiO_2_/PMHOS solution. In this manner, a superhydrophobic surface with a WCA of 156° and an SA of 6° was fabricated on wood substrates. The obtained superhydrophobic wood maintained the hydrophobicity even after several types of mechanical damage, including finger wiping, tape peeling, and so on. It also maintained its superhydrophobic performance after being dipped in aqueous solutions with a pH ranging from 2 to 8. More encouragingly, the resulting wood had good anti-icing and self-cleaning properties.

## 2. Materials and Methods

### 2.1. Materials

For this study, the wood materials (Chinese *Cunninghamia lanceolata* with density of 360 ± 50 kg/m^3^) were obtained from the Jiangsu province of China. The dimensions of the wood sample were 20 mm × 20 mm × 12 mm. The moisture content of the wood was 10%. The 1.5% PMHS was provided by Chenguang Research Institute of Chemical Industry (Chengdu, China). The ethanol (analytical grade) was from Tianjin Zhiyuan Chemical Reagent Co., Ltd (Tianjin, China). Tetraethoxysilane (TEOS) was purchased from Kermel (Tianjin, China). Sodium hydroxide was provided by Xilong Chemical Co., Ltd (Guangdong, China). Ammonium hydroxide solution was obtained from Sinopharm Chemical Reagent Co., Ltd (Shanghai, China). Liquids that are consumed daily, such as coffee, milk, soy sauce, and juice, were purchased from the local supermarket.

### 2.2. Fabrication of the Superhydrophobic PMHOS

First, 4 g PMHS was dispersed in a glass filled with 40 g ethanol; then, sodium hydroxide solution (0.2 g in 10 g of deionized water) was added. After the solution was magnetically stirred for 24 h and then aged for 1 day, a hydrolyzed gel was obtained. Then, hydrolyzed gel was dried at 120 °C for 4 h. The dried product was ground into powder using a mortar. To obtained PMHS powder with a smaller particle size, high-energy ball milling was performed at a speed of 600 rpm at room temperature using a frequency-conversion planetary grinder (BXQM-2L). The milling duration was 3 h. After high-energy grinding, the superhydrophobic PMHS powder, which was called PMHOS, could be better dispersed in the ethanol solution.

### 2.3. Synthesis of the SiO_2_/PMHOS Solution

A total of 16 g of TEOS was added into 50 g of ethanol and 1 g of PMHOS. Subsequently, the obtained solution was placed in ice water on an ultrasonic cell disruptor and sonicated for 30 min. A total of 1 g of ammonium hydroxide solution was added dropwise to the solution, stirred for 2 h, and stored to produce the SiO_2_/PMHOS solution by storing at room temperature for different numbers of days (0, 1, 2, 3, 4, and 5 days).

### 2.4. Surface Modification of the Wood Samples

The superhydrophobic wood was obtained by dipping the wood samples in the SiO_2_/PMHOS solution for 5 min. In order to prepare superhydrophobic wood with better durability, the wood samples were cyclically immersed in the solution for three cycles. Finally, all of the modified wood samples were heat-treated at 100 °C for 1 h.

### 2.5. Characterization

Observation of the micromorphological surface changes before and after the wood modification was performed by using scanning electron microscopy (SEM, ZEISS Z500). Fourier transform infrared spectroscopy (FTIR, Bruker Tensor 27) was used to characterize the functional group composition of the treated wood. The wood’s water contact angle (WCA) and sliding angle (SA) were measured on a contact angle meter (HARKE-SPCA-1, Beijing, China). The details of the testing process for WCA and SA were based on previous research reports [5,21]. The final WCA and SA were calculated by averaging five different positions on each sample.

## 3. Results and Discussion

### 3.1. Fabrication of Superhydrophobic Wood

The changes in the surface morphology and chemical composition of the wood before and after the treatment were determined with an SEM-EDS test. As shown in Figure 1a, the surface of the untreated wood had a large number of pits, and the surface was relatively smooth. Figure 1b shows an SEM image of a sample treated with the SiO_2_/PMHOS solution. Obviously, the modified wood surface was covered with abundant SiO_2_/PMHOS particles. Therefore, the surface roughness of the SiO_2_/PMHOS-modified wood was greater than that of the unmodified wood. The EDS results are presented as an inset in Figure 1a,b. The EDS analysis of the unmodified wood revealed that it contained C (80.92%) and O (19.08%) atoms, and the Si element was not detected. In contrast, the Si element appeared in the SiO_2_/PMHOS-modified wood from 0 to 12.72%. In conclusion, the SEM-EDS results indicate that the SiO_2_/PMHOS coating was successfully deposited on the wood surface.

The aging time of the SiO_2_/PMHOS solution had a significant effect on the wettability of the as-prepared wood surfaces. A series of wood samples were prepared using SiO_2_/PMHOS solutions with different aging times. As shown in Figure 2, when the aging time of the modifier solution increased from 0 to 3 d, the WCA of the modified wood surface increased from 134.9° to 153.1°. This shows that a superhydrophobic surface was obtained when the modifier solution was aged for a longer time. Consequently, the aging time had a decisive effect on the preparation of the superhydrophobic surfaces. In addition, after the modifier solution was aged for more than 3 d, the wood obtained through the modification was superhydrophobic.

### 3.2. Fourier Transform Infrared (FTIR) Characterization

FTIR was applied to characterize the chemical composition of the wood before and after the SiO_2_/PMHOS modification. As shown in Figure 3a, for the original wood, the band at 3419 cm^−1^ was assigned to the stretching vibration of the -OH groups [3,34], and the absorption band at 2926 cm^−1^ was attributed to the stretching vibration of the -CH_3_ groups [34]. In addition, the absorption peaks at 1735 cm^−1^ were due to the C=O stretching vibration [1]. In contrast, new absorption peaks appeared in the SiO_2_/PMHOS-modified wood samples, as shown in Figure 3b. The peaks at 2980 and 1273 cm^−1^ belonged to the stretching vibrations of -Si-CH_3_ of the PMHOS [35]. The absorption bands at 1094, 796, and 465 cm^−1^ were attributed to the characteristic Si-O-Si absorption bands [2,21]. In addition, the absorption band at 972 cm^−1^ was attributed to the -Si-OH stretching vibrations of the silica [36]. Therefore, the FTIR spectra confirm that the surface of the modified wood was covered with abundant SiO_2_/PMHOS particles.

### 3.3. The Durability of the Superhydrophobic Wood

To demonstrate the practicality of the SiO_2_/PMHOS-modified wood, the chemical stability of the SiO_2_/PMHOS-treated wood was characterized by immersing the wood samples in aqueous solutions at various pH values. As shown in Figure 4, superhydrophobic wood samples were immersed in corrosive solutions with pH values of 2, 4, 6, and 8. Every 30 min, the wood samples were taken out to measure their WCA and SA values. It was found that all samples had good hydrophobic properties, even after 120 min of immersion in the pH solutions (pH = 2, 4, 6, 8). However, the resistance of the superhydrophobic wood to a strong alkaline solution was relatively poor. After being immersed in an alkaline solution with a pH value of 13 for several minutes, the solution turned yellow. This indicated that the SiO_2_/PMHOS layer was destroyed, and then the lignin of wood dissolved into the solution. This was because the Si-O-Si bonds in the SiO_2_/PMHOS layer were easily hydrolyzed in the alkaline solutions. Fortunately, wood is generally exposed to acidic conditions, such as acid rain, in an outdoor environment. In addition, the stability of the SiO_2_/PMHOS layer on the wood surface against household pollutants, such as juice, milk, soy sauce, and coffee, was found to be good. Therefore, the poor resistance of SiO_2_/PMHOS-treated wood to an alkaline solution will not limit its practical application.

Abrasion tests were carried out to evaluate the mechanical stability of the superhydrophobic wood samples. As shown in Figure 5, the superhydrophobic wood samples were treated with a series of mechanical damages, including finger wiping (pressing with a finger for 10 s), knife scratching (scratching with a knife 10 times), tape peeling, and sandpaper abrasion (abrasion length of about 40 cm). The WCA and SA values of the resultant surfaces were measured after every abrasion. The results showed that the damaged wood surfaces retained their superhydrophobicity with a WCA of 150.3° and an SA of 9.4°. In addition, a falling sand test was also carried out to evaluate the mechanical stability of the SiO_2_/PMHOS-treated wood. As shown in Figure 6a, 100 g of sand grains (diameter: 100 to 200 μm) were allowed to fall from a height of 40 cm and hit the surface of the SiO_2_/PMHOS-modified wood. Then, the WCA was measured to estimate the mechanical stability of the superhydrophobic coating. As shown in Figure 6b,c, it was evident that the SiO_2_/PMHOS-modified wood surfaces remained superhydrophobic with a WCA of 152.4° after the falling sand test. These results indicated that the SiO_2_/PMHOS-modified wood surface had good abrasion resistance.

### 3.4. Multifunctional Wood Treated with SiO_2_/PMHOS

#### 3.4.1. Self-Cleaning and Anti-Fouling Property

Superhydrophobic wood is endowed with self-cleaning properties [26]. Consequently, water droplets readily roll off the surface and simultaneously carry away the contaminants accumulated around the surface. As shown in Figure 7, methylene blue powder was used to mimic contamination. Water was poured onto the surface to clean the polluted surface of the original and SiO_2_/PMHOS-modified wood. The water droplets dissolved the methylene blue for the original wood and then spread it on the wood surface. Finally, the wood surface was dyed blue, as shown in Figure 7a_1_–a_4_. As the water droplets were being poured onto the SiO_2_/PMHOS-modified wood, the water droplets rolled off the superhydrophobic surface. The methylene blue powder was removed by being dissolved into the water droplets. After the cleaning process, the surface of SiO_2_/PMHOS-modified wood was not stained by methylene blue, as shown in Figure 7b_1_–b_4_.

In addition to self-cleaning properties, the SiO_2_/PMHOS-modified wood also had excellent anti-fouling properties. As shown in Appendix A, the modified wood exhibited superhydrophobicity against common liquids, including soy sauce, milk, juice, and coffee. All of the liquid droplets had a spherical shape and were well supported by the wood surface treated with the SiO_2_/PMHOS solution, indicating good repellency towards these common liquids.

#### 3.4.2. Anti-Icing Property

Icing problems could trigger serious threats in many circumstances. Surface icing can result in corrosion and strength reduction, which largely decrease the service period of the material. Herein, the SiO_2_/PMHOS-modified wood samples had good anti-icing properties. As shown in Figure 8, anti-icing tests were conducted on the superhydrophobic wood samples. Briefly, the superhydrophobic wood samples were first soaked in water and then put into a −20 °C environment to freeze them; after 24 h of freezing, they were taken out. After the ice melted into water, the WCA of the wood was measured. It was evident that the SiO_2_/PMHOS-treated wood surfaces remained superhydrophobic with a WCA greater than 150° after the icing test. These results indicated that the SiO_2_/PMHOS-treated wood surface had good anti-icing properties.

### 3.5. Water Repellency and Dimensional Stability

To characterize the water repellency of the SiO_2_/PMHOS-modified wood, a moisture absorption test (24 h of immersion in water) was performed, as shown in Figure 9a. Obviously, the water absorption of the SiO_2_/PMHOS-modified wood was much lower than that of the unmodified wood (78.6%). When the aging time of the SiO_2_/PMHOS solution was 7 d, the resulting wood had low water absorption (32.62%). To evaluate the dimensional stability of the SiO_2_/PMHOS-modified wood, anti-swell efficiency (ASE) tests were conducted. When the value of the ASE exceeded 60%, it indicated that it had good dimensional stability [34]. The ASE values of the SiO_2_/PMHOS-modified wood are shown in Figure 9a. When the aging time of the SiO_2_/PMHOS solution was 5 or 7 d, the ASE values of the SiO_2_/PMHOS-treated wood were higher than 80%, indicating that the SiO_2_/PMHOS-modified wood had excellent dimensional stability.

In addition, the water uptake and the WCA of the SiO_2_/PMHOS-modified wood samples were also characterized after being placed in an indoor environment for 30 d, as shown in Figure 9b. The water absorption rate of the original wood in an indoor environment was 16.8%, while that of the SiO_2_/PMHOS-modified wood was only 4.6%. Moreover, the SiO_2_/PMHOS-modified wood still had superhydrophobic properties after being placed in an indoor environment for 30 d. These results indicated that the SiO_2_/PMHOS-modified wood had good water repellency.

### 3.6. The Formation Mechanism of the Superhydrophobic Wood Surface

Based on the results of the experiments in this study, the possible reaction mechanism of the fabricated superhydrophobic surface on wood via the method mentioned above can be summarized below (illustration shown in Figure 10). Particles of superhydrophobic PMHOS powder were fabricated, as shown in Figure 10b. PMHS contains many hydrophobic –CH_3_ groups and –Si–H groups. The –Si–H groups of PMHS possess ultra-high reactivity with hydroxyl (–OH) groups in the presence of a catalyst [5,36]. Ethanol contains –OH groups. Under base catalysis, it can undergo a dehydrogenation reaction with the –Si–H bond on the PMHS chain, thereby grafting the hydrophobic –OCH_2_CH_3_ (–OEt) group onto the PMHS chain and forming a hydrophobic sol [20]. After the sol was dried and ball milled, superhydrophobic PMHOS powder particles were obtained (as shown in Appendix A). The –Si–H bond in the PMHS chain was completely replaced by the hydrophobic –OEt groups. Then, the SiO_2_/PMHOS solution was prepared under alkaline conditions, and the four ethoxy groups (–OEt) on TEOS were hydrolyzed completely. The hydrolysate was present as Si(OH)_4_. After polycondensation of the hydrolysate, smaller secondary SiO_2_ particles were formed. These secondary particles aggregated with each other during the aging of the solvent to form larger particles. When the SiO_2_ particles grew to a certain size, due to the repulsive effect of the surface charge of the particles, it was difficult for the particles to re-aggregate, and a stable sol was obtained, as shown in Figure 10a. In addition, because the superhydrophobic PMHOS powder particles were added before the sol was formed, the resulting sol was the SiO_2_/PMHOS solution. Then, the superhydrophobic surface was constructed on the surface of the wood by soaking the original wood in the SiO_2_/PMHOS solution, as shown in Figure 10c. The formation of a superhydrophobic layer on the wood’s surface can be attributed to the formation of a rough layer of SiO_2_/PMHOS coatings on the surface. PMHOS is a product of organic polymerization, which can be considered a low-surface-energy substance. The surface of SiO_2_ contained a large number of -OH groups, which could react with the –OH groups on the surface of the wood. In other words, by soaking the wood with the SiO_2_/PMHOS solution, not only could the surface energy of the wood be reduced, but the surface roughness could also be increased. This indicated that the SiO_2_/PMHOS modification reached the necessary conditions for forming a superhydrophobic surface.

## 4. Conclusions

In summary, superhydrophobic wood with good durability was successfully fabricated by simply soaking wood in a SiO_2_/PMHOS solution. A superhydrophobic state was achieved in the wood following the aging of the SiO_2_/PMHOS solution and immersion of the wood. The proposed formation mechanism of superhydrophobicity on the wood surface resulted from PMHOS, a product of organic polymerization with low surface energy, and the formation of a rough layer of SiO_2_/PMHOS coatings on the surface of the wood. The superhydrophobic surface possessed good durability after sustaining mechanical damage, including damage from finger wiping, tape peeling, knife scratching, and sandpaper abrasion. In addition, the superhydrophobic surface had self-cleaning properties, chemical stability, improved water resistance, dimensional stability, and good anti-icing properties. These excellent properties can expand the potential and scope of applications of the superhydrophobic wood prepared in this study.

## Figures and Tables

**Figure 1 polymers-14-01953-f001:**
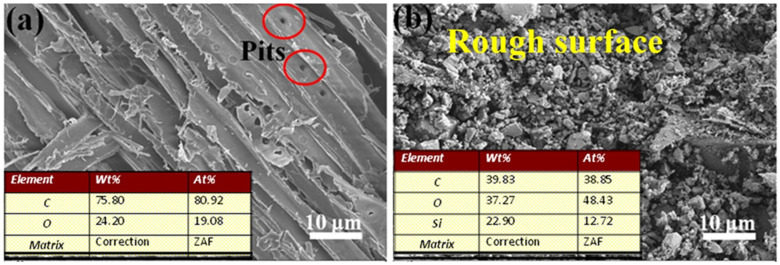
Surface morphology and EDS analysis of the wood substrate before (**a**) and after (**b**) the superhydrophobic treatment.

**Figure 2 polymers-14-01953-f002:**
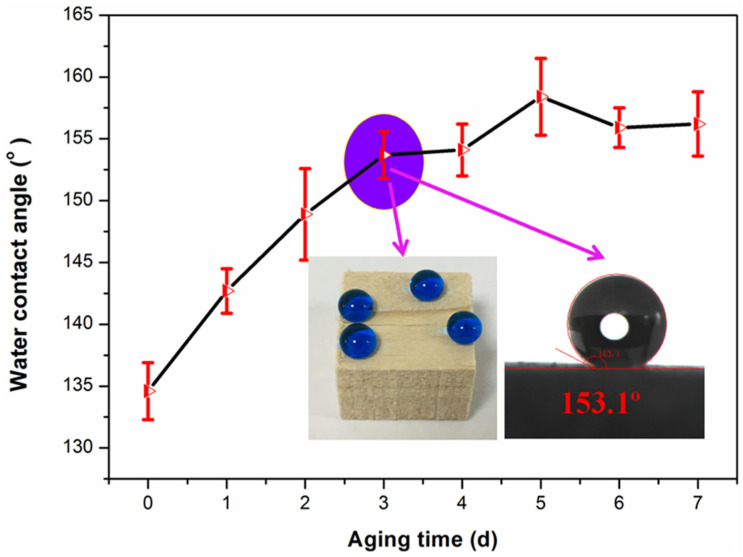
WCA (radial section) on the surfaces of modified wood samples with different aging times of the SiO_2_/PMHOS solution.

**Figure 3 polymers-14-01953-f003:**
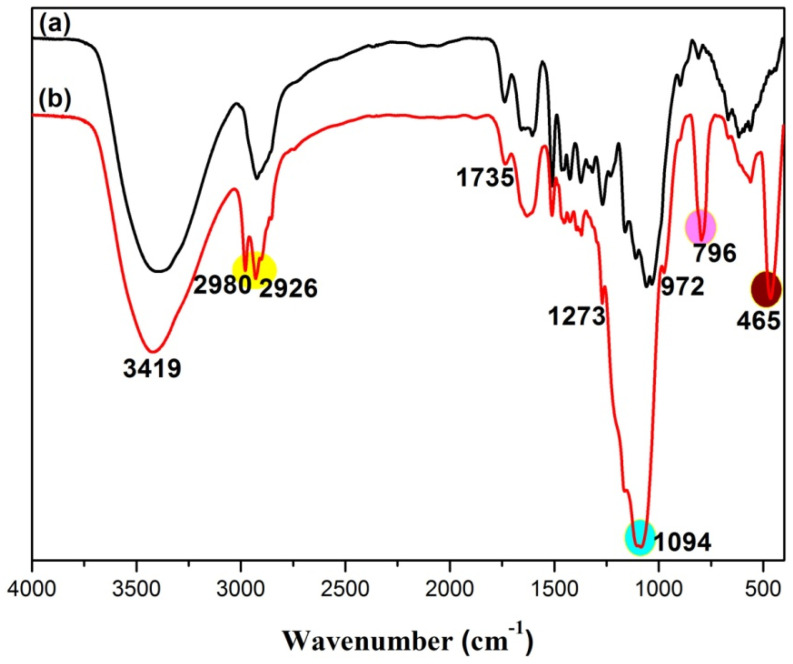
FTIR spectra of the original wood sample and the wood sample treated with the SiO_2_/PMHOS solution ((**a**) original wood; (**b**) modified wood).

**Figure 4 polymers-14-01953-f004:**
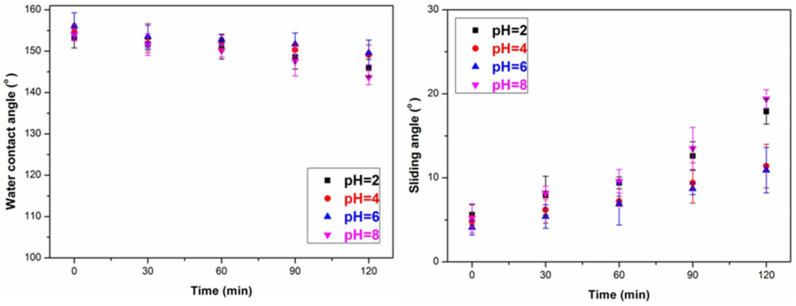
Variations in the WCA and SA in solutions with different pH values.

**Figure 5 polymers-14-01953-f005:**
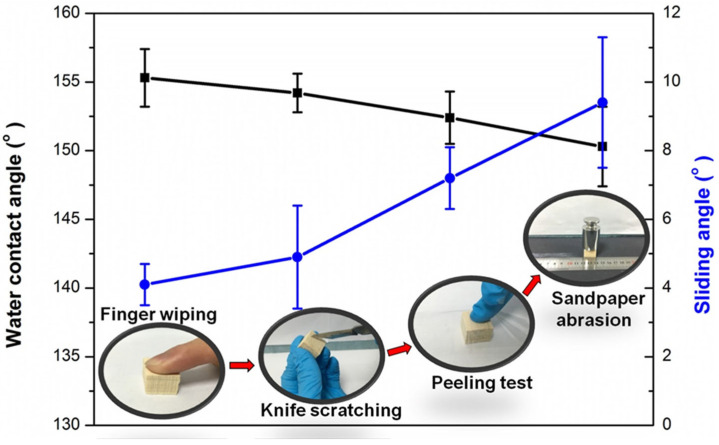
Changes in the WCA and SA during a sequential abrasions, including finger wiping, knife scratching, tape peeling, and sandpaper abrasion.

**Figure 6 polymers-14-01953-f006:**
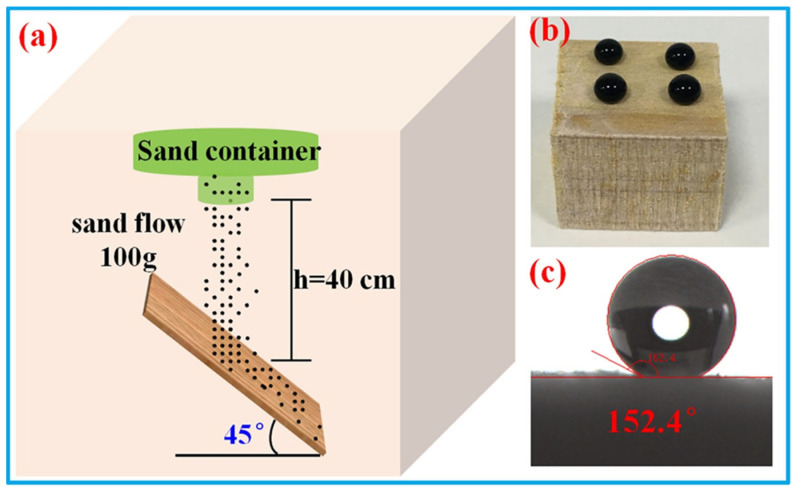
Schematic diagram of the falling sand impact abrasion tests (**a**); a water drop deposited on the as-prepared wood surface after sand abrasion (**b**) and the WCA test of wood surface after sand abrasion (**c**).

**Figure 7 polymers-14-01953-f007:**
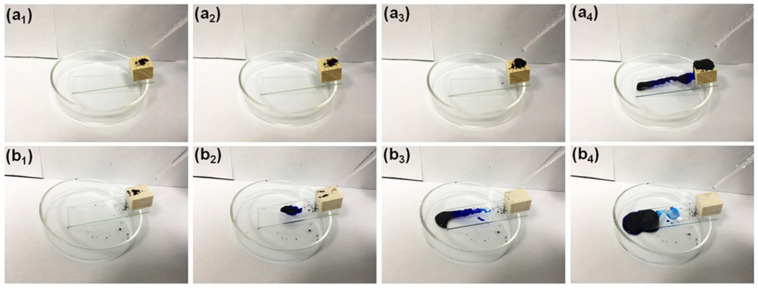
Wood self-cleaning test before (**a_1_**–**a_4_**) and after the SiO_2_/PMHOS-modified (**b_1_**–**b_4_**).

**Figure 8 polymers-14-01953-f008:**
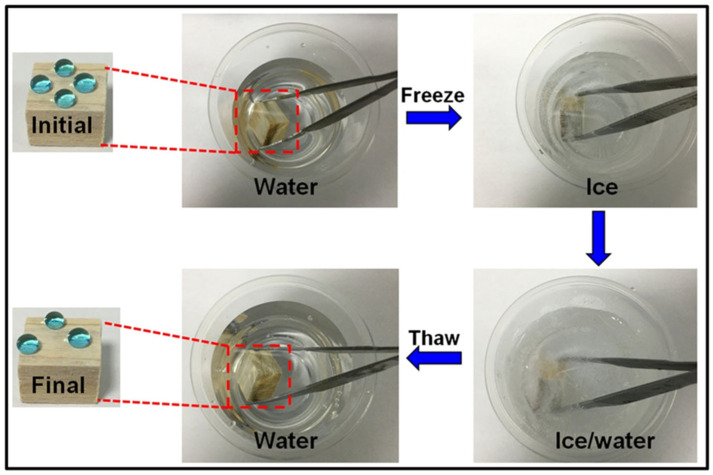
Anti-icing test of the SiO_2_/PMHOS-treated wood.

**Figure 9 polymers-14-01953-f009:**
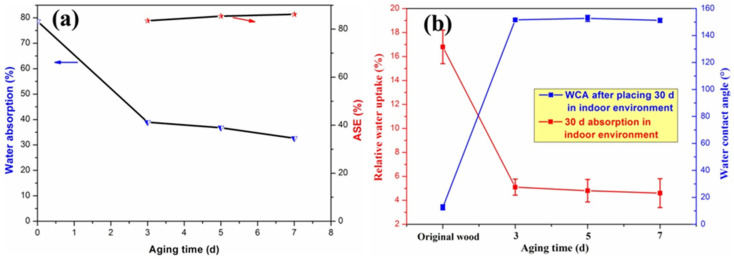
(**a**) The tests of 24 h of water absorption and the ASE; (**b**) 30 d water absorption rate and WCA value of the original and SiO_2_/PMHOS-modified wood in an indoor environment.

**Figure 10 polymers-14-01953-f010:**
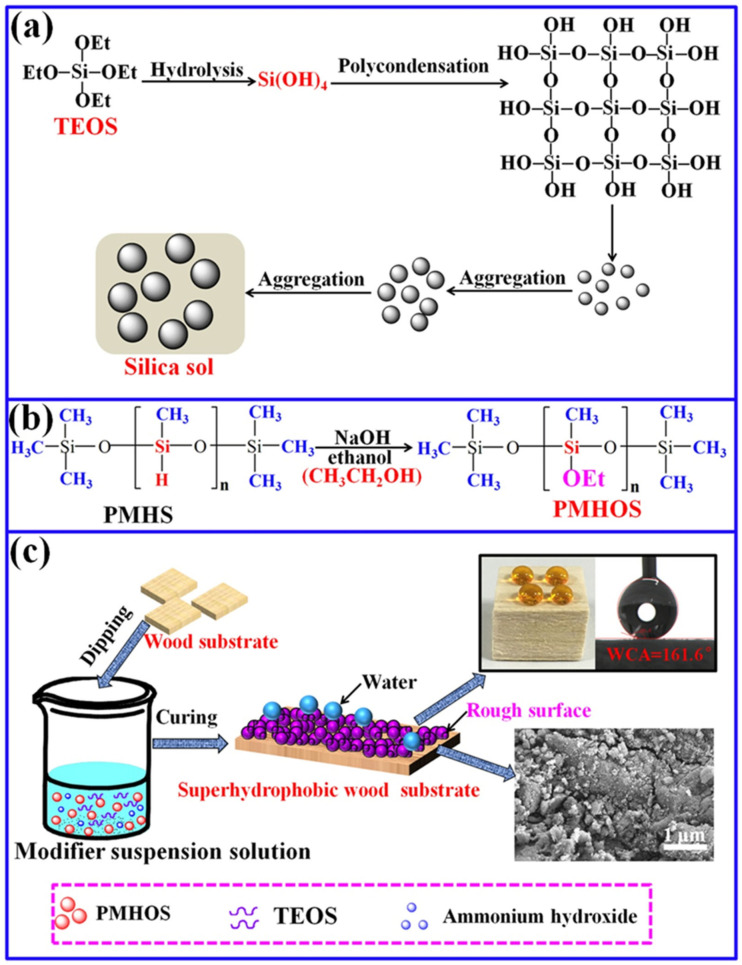
Schematic illustration of the formation mechanism of the superhydrophobic wood surface: (**a**) the formation process of silica sol; (**b**) the preparation mechanism of PMHOS powder; (**c**) the preparation of superhydrophobic wood.

## Data Availability

The data presented in this study are available on request from the corresponding author.

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
