# Peer review of "Facile Fabrication of Fluorine-Free, Anti-Icing, and Multifunctional Superhydrophobic Surface on Wood Substrates"

_polymers, 2022, doi:10.3390/polym14101953_

Round 1

Reviewer 1 Report

In this study multifunctional superhydrophobic wood was successfully fabricated by introducing SiO2 sol and superhydrophobic powder (PMHOS). The topic is interested for the readers but Authors should make changes and additions into the text.

Keywords:

  • The word "wettability" to the keywords should be added.
  • Authors should consider addition of another keywords strongly connected with the topic.

Introduction

  • More papers related to the wettability should be given (e.g. Authors). Readers would like to know experience of Authors.

Materials and methods

  • More information about properties of Chinese Cunninghamia lanceolata wood specie should be given (density, moisture content).
  • How many samples were used for the investigations?
  • What about conditioning parameters in the laboratory (temperature, relative humidity)?
  • Why Authors used such liquids (coffee, milk, soy sauce, juice, and beer) for the contact angle measurements?
  • How many measurements of the contact angle were performed?

Results

  • Authors used many investigations without explanation. For example on the Fig. 5  Changes in water contact angles and sliding angles during a sequential abrasion damage, including finger wiping, knife scratching, tape peeling, and sandpaper abrasion were given. The reviewer can not find description of this methods in the reviewed paper. It should be underlined interesting idea, but it can be a topic of other paper.
  • Fig. 10 is not necessary or need more additional comments.

Conclusions:

  • The conclusions are formulated in a general way. This chapter needs to be rewritten.

References:

  • Authors cited many papers of scientists from Asia. Is the topic popular only in such region? Please explain it.
  • References are numbered twice.

I recommend the paper for the publishing after major changes and additions.

Author Response

Response to Reviewer 1 Comments

Point 1: The word "wettability" to the keywords should be added.

Response 1: Thank you very much for your suggestion. We have added it in the new submitted manuscript..

Point 2: Authors should consider addition of another keywords strongly connected with the topic.

Response 2: Thank you very much for your suggestion. We have corrected it in the new submitted manuscript.

Point 3: More papers related to the wettability should be given (e.g. Authors). Readers would like to know experience of Authors.

Response 3: Thank you very much for your suggestion. We have corrected it in the new submitted manuscript.

Point 4: More information about properties of Chinese Cunninghamia lanceolata wood specie should be given (density, moisture content).

Response 4: Thank you very much for your suggestion. We have added it in the new submitted manuscript.

Point 5: How many samples were used for the investigations?

Response 5: OK. Except for SEM-EDS and FTIR tests, 5 wood samples were used for each parallel experiment. Approximately 150 wood samples were used in this study.

Point 6: What about conditioning parameters in the laboratory (temperature, relative humidity)?

Response 6: OK. During this study, the laboratory temperature was maintained at room temperature and the relative humidity was maintained at 61%.

Point 7: Why Authors used such liquids (coffee, milk, soy sauce, juice, and beer) for the contact angle measurements?

Response 7: We deeply appreciate the reviewer's suggestion and comment. The contact angle test with these liquids is because wood is mainly contaminated by these liquids in daily use.

Point 8: How many measurements of the contact angle were performed?

Response 8: Thank you very much for your suggestion. The WCA and SA values reported are the mean of 5 measurements. This part of the information is provided in the characterization

Point 9: Authors used many investigations without explanation. For example on the Fig. 5 Changes in water contact angles and sliding angles during a sequential abrasion damage, including finger wiping, knife scratching, tape peeling, and sandpaper abrasion were given. The reviewer can not find description of this methods in the reviewed paper. It should be underlined interesting idea, but it can be a topic of other paper.

Response 9: Thank you very much for your suggestion. We have added these details in the new submitted manuscript.

Point 10: Fig. 10 is not necessary or need more additional comments.

Response 10: Thank you very much for your suggestion. We have corrected it in the new submitted manuscript.

Point 11: The conclusions are formulated in a general way. This chapter needs to be rewritten.

Response 11: Thank you very much for your suggestion. We have corrected it in the new submitted manuscript.

Point 12: Authors cited many papers of scientists from Asia. Is the topic popular only in such region? Please explain it.

Response 12: We deeply appreciate the reviewer's suggestion and comment. In fact, wood modification research is not only popular in Asia, there are also many scholars in Europe or other regions who have carried out modification research on wood. The reason why this article cites more research reports by Asian scholars is simply because we know more about their research.

Point 13: References are numbered twice.

Response 13: Thank you very much for your suggestion. We have corrected it in the new submitted manuscript.

Reviewer 2 Report

Dear Authors in attach your files with some comments of mine. In theory the study could be of some interest also if I am not sure polymersi is the right journal. There are on my opinion major flows. 

The main object of the article in introduction is missing.

Methods are very poor with some parts described directly in the results.

I do not agree totally with some description of your results and the interpretation...

I think the manuscript must be seriously improved to be worth of pubblication.

Author Response

Response to Reviewer 2 Comments

Point 1: The main object of the article in introduction is missing.

Response 1: Thank you very much for your suggestion. The main object of this article is about hydrophobic modification of wood. In the introduction part, we also pointed out the main object of the research of this article.

Point 2: Methods are very poor with some parts described directly in the results.

Response 2: We deeply appreciate the reviewer's suggestion and comment. We have corrected it in the new submitted manuscript.

Point 3: I do not agree totally with some description of your results and the interpretation.

Response 3: We deeply appreciate the reviewer's suggestion and comment. The new submitted manuscript was revised carefully.

Point 4: I think the manuscript must be seriously improved to be worth of publication.

Response 4: Thank you very much for your suggestion. The new submitted manuscript was revised carefully.

Round 2

Reviewer 1 Report

Authors made corrections and additions to the text, but did not consider into account all of the reviewer's comments.

  • The words "wettability" and "multifunctionality" correctly were added to the keywords in the submitted manuscript.
  • Information on density and moisture content of Chinese Cunninghamia lanceolata wood is provided. It would be better to give the density in another unit (kg/m3).
  • The authors used approximately 150 wood samples for the study. Why this number of samples?
  • The samples were conditioned at room temperature (please specify range?) and 61% relative humidity (constant or different?).
  • There is no standard or information on the selection of liquids (coffee, milk, soy sauce, juice and beer) used for the contact angle measurements.
  • Authors have added some information related to test methods.
  • The conclusions are still stated in general terms. This chapter needs to be rewritten.
  • Authors did not take into account the reviewer's suggestion and did not add publications of other researchers.
  • The numbering of the references was corrected.

Author Response

Response to Reviewer 1 Comments

We deeply appreciate the reviewer's suggestion and comment. Based on your comments and suggestions, we have made a lot of revisions in the new submitted manuscript.

Point 1: Authors made corrections and additions to the text, but did not consider into account all of the reviewer's comments..

Response 1: Sorry for our mistake. Based on your comments and suggestions, we have made a lot of revisions in the new submitted manuscript.

Point 2: The words "wettability" and "multifunctionality" correctly were added to the keywords in the submitted manuscript.

Response 2: Thank you very much for your suggestion. We have corrected it in the new submitted manuscript.

Point 3: Information on density and moisture content of Chinese Cunninghamia lanceolata wood is provided. It would be better to give the density in another unit (kg/m3).

Response 3: Thank you very much for your suggestion. We have corrected it in the new submitted manuscript.

Point 4: The authors used approximately 150 wood samples for the study. Why this number of samples?

Response 4: OK. This is due to an operational error during some characterization experiments, which resulted in the use of approximately 150 wood samples for this study.

Point 5: The samples were conditioned at room temperature (please specify range?) and 61% relative humidity (constant or different?).

Response 5: Thank you very much for your suggestion. The experimental temperature range was 25 ℃±2 ℃ and the relative humidity was kept at 61±3%.

Point 6: There is no standard or information on the selection of liquids (coffee, milk, soy sauce, juice and beer) used for the contact angle measurements.

Response 6: Thank you very much for your suggestion. We have added these details in the new submitted manuscript..

Point 7: Authors have added some information related to test methods.

Response 7: Thank you very much for your suggestion.

Point 8: The conclusions are still stated in general terms. This chapter needs to be rewritten.

Response 8: Thank you very much for your suggestion. We have corrected it in the new submitted manuscript.

Point 9: Authors did not take into account the reviewer's suggestion and did not add publications of other researchers.

Response 9: Sorry for our mistake. We have added these details in the new submitted manuscript.

Point 10: The numbering of the references was corrected.

Response 10: Thank you very much for your suggestion.
